# Prevalence of and Risk Factors for Iron Deficiency in Twin and Singleton Newborns

**DOI:** 10.3390/nu14183854

**Published:** 2022-09-17

**Authors:** Rebecca K. Campbell, Catalin S. Buhimschi, Guomao Zhao, Cielo Dela Rosa, Bethany T. Stetson, Carl H. Backes, Irina A. Buhimschi

**Affiliations:** 1Division of Epidemiology and Biostatistics, School of Public Health, University of Illinois Chicago, Chicago, IL 60612, USA; 2Department of Obstetrics and Gynecology, College of Medicine, University of Illinois Chicago, Chicago, IL 60612, USA; 3Department of Obstetrics and Gynecology, Northwestern University Feinberg School of Medicine, Chicago, IL 60611, USA; 4Center for Perinatal Research, The Abigail Wexner Research Institute at Nationwide Children’s Hospital, Columbus, OH 43215, USA; 5Department of Pediatrics, The Ohio State University College of Medicine, Columbus, OH 43215, USA

**Keywords:** iron deficiency, neurodevelopment, infancy, multiple gestation pregnancy, nutrition, pregnancy, fetal development, epidemiology

## Abstract

Iron deficiency (ID) in utero and in infancy can cause irreversible neurocognitive damage. Iron status is not routinely tested at birth, so the burden of neonatal ID in the United States is unknown. Infants born from twin or higher-order pregnancies may be at elevated risk of inadequate nutrient endowment at birth. The present study sought to compare the burden of neonatal ID in cord blood serum samples from twin (*n* = 54) and singleton pregnancies (*n* = 24). Iron status (serum ferritin (SF), soluble transferrin receptor (sTfR), hepcidin) and inflammation (C-reactive protein (CRP) and interleukin-6 (IL-6)) biomarker concentrations were measured by immunoassay. The prevalence of ID (SF < 76 ng/mL) among twins was 21% (23/108) and among singletons 20% (5/24). Gestational age at birth, maternal race and infant sex predicted SF levels. Maternal anemia (hemoglobin < 11 g/dL) was observed in 40% of mothers but was not associated with neonatal iron biomarkers. More research is needed to identify risk factors and regulatory mechanisms for inadequate fetal iron accrual to identify higher risk pregnancies and neonates for screening and intervention.

## 1. Introduction

Protecting children’s cognitive development is a global public health priority, as it lays the foundation for a healthy and productive life [1]. Iron (Fe) deficiency (ID) is highly prevalent worldwide, especially among young children and women of reproductive age [2], and is among the major causes of suboptimal childhood cognitive development [3]. In the US, an estimated 10% of women of reproductive age and 15% of toddlers have ID [4]. Iron status is not routinely tested in early infancy [5], and national surveillance via the National Health and Nutrition Examination Survey (NHANES) begins in the toddler years. Still, evidence is clear that early infancy ID is a strong risk factor for ID in later infancy and for impaired cognitive development [6,7,8,9]. Fe stores accrued in utero are essential for supporting infant and early childhood cognitive development, as cognitive deficits are only partially corrected with postnatal iron supplementation [6,7,8,9,10,11].

Twins comprise about 3% of births in the US at present, though they account for a much larger proportion of preterm and low birthweight births [12,13]. Infants born of twin or higher order pregnancies require particular attention with respect to their nutrient endowment and neurocognitive development [14]. Fetal Fe accrual accelerates in the third trimester of pregnancy, such that babies born preterm (≤37 weeks) are at greater risk of low Fe stores at birth and of ID later in infancy [10]. Twins and higher order multiples are at greatly increased risk of preterm birth, and preterm infants from multiple gestation pregnancies tend to be at greatest risk of impaired cognitive development [15]. The extent to which this is due to preterm birth or to other factors, such as the greater nutritional demands of multiple gestation pregnancies, is not clear [14,16]. Evidence regarding iron stores in twin fetuses and iron partitioning among mother and fetuses in multiple gestation pregnancies is sparse. 

The US burden of ID is unequal between subpopulations defined by race and ethnicity: Black and Latinx toddlers and women are at highest risk of ID and anemia [4,17,18]. These disparities are not well explained, though some studies suggest obesity, diet, and child feeding practices may contribute [17,19,20,21]. Racial disparities in ID may particularly affect twin neonates, as spontaneous twinning and spontaneous preterm birth are more common in Black women in the US [12,22,23,24]. The extent to which racial/ethnic disparities in ID are transmitted in utero from mother to infant is also not known. 

Evidence regarding neonatal ID burden and risk factors in twin pregnancies is needed to support screening and prenatal interventions that protect early childhood cognitive development. The present study is a pilot that aimed to assess cord blood Fe status and its determinants in twins and in a comparator group of singletons enrolled in the same geographic area in Columbus, Ohio (USA). We hypothesized that twins would have a high prevalence of ID at birth compared to singletons and that sociodemographic and clinical characteristics would identify higher-risk pregnancies. 

## 2. Materials and Methods

The present study aimed to evaluate fetal iron status at birth in twin versus singletons. Pregnant persons receiving care at The Ohio State University Wexner Medical Center (OSUWMC, Columbus, OH, USA) were enrolled in the antepartum clinic and Labor and Birth Unit from August 2014 to November 2017. All subjects signed informed consent for collection of biological samples and medical chart data abstraction. Using a large gauge needle to prevent hemolysis, venous cord blood was collected in sterile fashion, within minutes after birth. The blood was allowed to clot, centrifuged and serum collected, aliquoted and stored at −80 °C. The biorepository protocols were approved by the Institutional Review Boards of OSUWMC and The Research Institute at Nationwide Children’s Hospital (# 2013H0404, 22 January 2014). The study was reviewed and determined “not human subject research” by the Office for the Protection of Research Subjects at University of Illinois Chicago (# 2021-0628, 10 June 2021) where the laboratory assays on de-identified specimens were conducted in 2021. 

For the present study, cord blood samples from 108 neonates from 54 twin pregnancies (47 diamniotic—dichorionic (“di-di”); 7 monochorionic-diamniotic (“mono-di”)) and 24 neonates from singleton pregnancies were selected from the biorepository. A sample size of at least 100 twin neonates was needed to allow for detecting a linear association between the continuous iron biomarkers and neonate characteristics at β = 0.20 and α = 0.05, with a Bonferroni correction for multiple (15) planned comparisons and variance inflation factor (VIF) = 1.47 [25]. Twin pregnancies were selected from the biorepository participants without restricting by delivery context or complications typical of twin pregnancies (e.g., spontaneous preterm labor, preeclampsia). Singletons were selected to have a similar representation of delivery context, pregnancy complications and basic metadata (e.g., infant sex) compared to the twins. Infants with major congenital anomalies were excluded from the present study. Individual samples were not matched on gestational age or other characteristics between twins and singletons.

Concentrations of serum ferritin (SF), soluble transferrin receptor (sTfR), hepcidin, C-reactive protein (CRP) and interleukin-6 (IL-6) were determined with commercial immunoassays kits (Appendix A). In a small number of samples (*n* = 3), only plasma but not serum was available for one neonate of a twin pair, so assays were run using plasma. SF and sTfR are markers of iron stores [26], while hepcidin is a hormone produced in the liver that regulates iron absorption and storage [27]. SF, sTfR and hepcidin are all acute phase reactants, so they must be measured in parallel with biomarkers of inflammation (e.g., CRP and IL-6), to aid in their interpretation [28]. 

Fetal ID was defined as cord blood SF <76 ng/dL [29,30]. Fetal inflammation was defined as cord blood IL-6 ≥100 pg/mL [31,32]. No samples had CRP values that exceeded the typical cutoff of 5 mg/L [33]. The 95th percentile of the distribution of CRP values was used instead as the cutoff for inflammation as defined by CRP. 

Maternal sociodemographic and health data were collected from women at enrollment and from their electronic medical records. Maternal race was categorized as White, Black or other. Ethnicity was dichotomized as Hispanic or not Hispanic. Maternal age was grouped into <25 years, 25–34 years and ≥35 years. Gravidity was dichotomized 1 or 2+; parity was dichotomized 0 or 1+. Hemoglobin (Hb) in late pregnancy (3rd trimester or at delivery) was available for about 2/3 of participating women. Maternal prenatal anemia was defined as Hb <11 g/dL [34]. Height and weight in late pregnancy to calculate body mass index (BMI) were similarly available for many but not all of the mothers. Gestational age at birth, delivery type, birth weight and infant sex were ascertained from the medical record. Chronicity (mono-di or di-di) was confirmed in all cases through pathologic examination of the placenta. None of the mono-chorionic twin gestations displayed clinical or ultrasonographic signs of twin-to-twin transfusion. Preterm birth was defined as birth at <37 weeks’ gestation. Low birthweight (LBW) was <2500 g and very low birthweight (VLBW) <1500 g. 

### Statistical Analysis

Analyses began with descriptive analyses. Differences in maternal and infant characteristics between twins and singletons in the sample were evaluated with χ2 tests and t-test for dichotomous and continuous characteristics, respectively. χ2 tests and t-tests for maternal characteristics were conducted at the mother level not the infant level to avoid double-counting mothers of twins. Distributions of biomarker values were examined for normality and log-transformed if needed. Mixed-effects linear regression models with a random effect for twin pair were used to summarize biomarker central tendency and variability in twins and bivariate associations among biomarkers. For singletons, simple linear regression models were used to summarize sample means and standard errors. 

To evaluate maternal, pregnancy and infant correlates of ID and of continuous iron biomarkers, mixed effects logistic and linear regression models were developed with random effects for twin pairs. Predictors were evaluated individually in simple linear and logistic models. As multiple potentially interrelated characteristics were found to be associated with SF in univariate models, a multivariable model was developed mutually adjusted for the statistically significant characteristics in the univariate models.

Associations between pairs of biomarkers and with gestational age and birthweight were calculated in unadjusted mixed effects linear regression models to allow for non-independence of observations within twin pairs. 

Concordance of biomarkers within twin pairs was examined with intraclass correlation coefficients (ICCs), a measure of the extent to which the total variability in a variable is explained by between-cluster differences. ICCs were calculated as between-cluster variance in mixed effects linear regression models. To describe predictors of intrauterine (within twin pair) variability in iron biomarkers, intracluster variance (σ_u_) was compared between mixed effects models without and with each individual predictor. 

Finally, a sensitivity analysis for the impact of including biomarker values from plasma samples was conducted. The models described above were rerun (1) omitting *n* = 3 plasma samples and (2) adjusting analytes for total protein in the sample. Neither qualitatively impacted the findings; final models include plasma samples and no adjustment for total protein. Analyses were conducted in Stata version 16.1 (StataCorp LLC., College Station, TX, USA).

## 3. Results

Biomarker data were available for 132 neonates from 78 pregnancies: 24 from singleton pregnancies and 108 from twin pregnancies. In the twin group, mothers were of mean age 28.9 years, 78% identified as White and 96% as non-Hispanic (Table 1). More than half (54%) were primiparous and 46% had anemia in late pregnancy. For twin gestations the mean gestational age at birth was 35 weeks and mean birthweight was 2239 g, with 64% of neonates classified as low birthweight. Most twins (87%) were dichorionic- diamniotic. Singletons in the study had similar maternal and infant characteristics to twins, though a higher proportion of twins were born with low birthweight. 

Overall, there was no significant difference in cord blood ferritin levels among twin vs. singleton neonates. Approximately 21% of twin (23/108) and singleton (5/24) neonates had iron deficiency (SF <76 ng/mL) (Table 2). Iron biomarkers suggested slightly better iron status in twins compared to singletons: SF and hepcidin were higher and mean sTfR was lower in twins compared to singletons. CRP and IL-6 were infrequently elevated such that differences between singletons and twins could not be evaluated statistically, though more twins had elevated inflammation markers.

Twins and singletons had similar prevalence of ID. Neonates of Black mothers had approximately double the prevalence of ID compared to White mothers (39% vs. 17%) (Figure 1). Neonates of mothers with anemia and those born preterm also had higher prevalence of ID. Male and female neonates had similar rates of ID. In mixed effects logistic regression analyses adjusted for clustering within twin pairs, no maternal or infant characteristics were associated with ID prevalence (Appendix A). 

Iron biomarkers largely demonstrated expected bivariate associations (Figure 2). sTfR, which is present in circulation in proportion to tissue demand for iron and is thus elevated when iron supply is low, was inversely associated with both SF and hepcidin and positively associated with the inflammation markers CRP and IL-6. Hepcidin, which suppresses iron absorption and circulation, was positively associated with SF and with CRP, IL-6 and gestational age at birth. Of the iron and inflammation biomarkers, only IL-6 was associated with birthweight, with higher IL-6 in babies with lower birthweight.

In mixed effects linear regression models with each continuous iron biomarker as the dependent variable, SF was lower in neonates of Black vs. White mothers, male vs. female neonates and in those born preterm compared to term (Appendix A). sTfR was higher in neonates of Black mothers compared to neonates of White mothers and lower in twins compared to singletons. Hepcidin was greater with increasing gestational age and lower in neonates born preterm compared to term. Predictors of neonatal SF, sTfR and hepcidin were independent of maternal anemia. In a multivariable mixed effects model with SF as the dependent variable, maternal race, preterm birth and infant sex were each independent predictors of SF (Table 3). 

ICCs describe the degree to which variability in a marker is explained by differences between clusters (twin pairs, in this case). ICCs ranged from 0.75 for SF to 0.26 for IL-6 (Figure 3). SF, CRP and sTfR all had ICCs well above 0.5, indicating most variability was between twin pairs. Hepcidin had ICC closer to 0.5, suggesting more equal contributions of variability within and between pairs, while most variability in IL-6 occurred within pairs. This is consistent with elevated IL-6 being uncommon and typically only affecting one baby in a twin pair. ICCs among mono-di twins were much closer to 1 for SF, sTfR, hepcidin and CRP compared to di-di twins, suggesting lower intrauterine variability in monochorionic pregnancies. Iron deficiency was largely concordant within twin pairs. Seventy percent of pairs both had sufficient iron stores, 13% were both iron deficient and 17% were discordant for iron status. Iron deficiency concordance did not differ significantly by chorionicity (not shown).

When maternal and infant characteristics were examined for the percent of intrauterine variability they explained, cord blood hepcidin explained nearly 25% of the intrauterine variability in SF and 15% of the intrauterine variability in sTfR (Table 4). For sTfR, chorionicity explained 25% of intrauterine variability, with much higher correlations in sTfR within twin pairs who shared a placenta compared to dichorionic pregnancies. Interestingly, chorionicity explained only about 1% of the intrauterine variability in SF and hepcidin. 

## 4. Discussion

In this sample of neonates from twin and singleton pregnancies, iron deficiency affected 20% of neonates in both groups. Concurrently, 40% of the mothers had anemia in late pregnancy. Together, this suggests substantial risk for pregnancy complications and impaired development outcomes stemming from inadequate iron status in an apparently healthy population. 

Current evidence on the burden of neonatal ID comes from small studies in suspected high-risk populations. Two studies in neonatal intensive care unit (NICU) patients in the Intermountain Health system in Utah reported 17% and 13% of neonates had ID in a pilot and subsequent cross-sectional study, respectively [35,36]. Notably, the pilot study included a low-risk reference group in which only one in 20 (5%) sampled neonates had ID [36]. The rate of ID in singletons in our study may have been higher due to a higher prevalence of preterm birth in our sample. Another study of babies born to adolescent mothers, also a higher risk group, reported 25% of neonates had ID [30]. In the present study, we did not select for maternal or infant risk factors aside from multiple gestation pregnancy, yet the observed prevalence of ID was similar to higher risk samples. 

The prevalence of ID we observed coupled with the known irreversible effects of neonatal ID [6,37,38,39,40,41,42,43,44] should instigate larger studies towards rapid and widespread action on this issue. Early infancy screening and expanded iron supplementation likely are needed to protect iron status and improve cognitive development in iron deficient neonates. In the longer term, identifying at-risk pregnancies and intervening prenatally should be prioritized to prevent neurocognitive impairments.

The available data allowed for only a partial examination of risk factors for neonatal ID and iron status biomarkers. None of the examined maternal and infant characteristics was associated with ID. Some characteristics predictive of SF were identified, while few of the examined characteristics were associated with sTfR or hepcidin. We saw that preterm birth is inversely associated with SF, and boys have lower SF in cord blood compared to girls, which concurs with prior evidence [10]. Determinants of iron status in twin neonates specifically are not well described in the literature. The extent to which gestational age is related to iron status in twins is unknown, but it is thought to differ from the relationship in singletons because of differing fetal growth trajectories and mechanisms for early onset of labor [45]. More detailed examination of spontaneous and provider-initiated preterm deliveries in a future study with a larger sample may be necessary to better understand the association of neonatal iron status with preterm birth in twins, especially given the complex role of inflammation in spontaneous preterm birth and iron status assessment. 

In the present study, we observed that neonates of Black mothers had lower mean SF and higher mean sTfR, which may be consistent with a prior study in newborns of adolescents that found lower Hb in neonates of Black compared to White mothers [30]. While Black women of reproductive age and Black toddlers are known to have higher rates of ID compared to white women and toddlers in the US [4,17,18], the extent to which racial disparities in ID are present at birth has not been examined. 

Hepcidin explained a substantial portion (25%) of intrauterine variability in SF, suggesting a role for hepcidin in fetal iron partitioning in twin pregnancy [45]. In a similar study, Ru et al. reported hepcidin explained 48% of SF intrauterine variability, but they also reported lower SF ICC (greater intrauterine variability) and lower mean SF [25]. Fetal hepcidin may explain relatively more variability in SF when SF levels are low and/or differ more within twin pairs. Hepcidin also explained a substantial but smaller portion (11.6%) of intrauterine variability in sTfR. Further research that includes maternal and placental measures or transport mechanism is needed to characterize regulation of iron partitioning in multiple gestation pregnancies. 

Somewhat in contrast to neonatal ID, anemia in pregnant and reproductive age women is a well-recognized and longstanding nutrition concern [46,47]. Still, the 40% prevalence of anemia in late pregnancy that we observed is much higher than what is typically reported in US settings [48]. It is, however, similar to the prevalence of anemia in another study of multiple gestation pregnancies [49], suggesting mothers pregnant with twins may be at particularly high risk of anemia. 

Despite high rates of maternal anemia, we did not observe associations between maternal anemia and neonatal iron status. This could be due to the timing of maternal Hb assessment: maternal Hb during labor may be transiently suppressed by inflammation related to labor rather than indicating iron deficiency anemia. Or, fetal iron could be protected at the expense of maternal iron status [10]. Maternal iron status and repletion measured in more detail longitudinally during pregnancy, along with examination of placental iron stores and regulatory markers, would help elucidate these dynamics. These data along with aspects of prenatal health and prenatal care would enhance future studies and help translate them into clinical interventions.

Strengths of the present study include the focus on neonates from twin pregnancies and the availability of cord blood samples, which is a major strength of this biorepository and allowed for assessment of a detailed iron panel including hepcidin. Prior analysis of inflammatory markers in the same samples allowed for inclusion of IL-6, which confirmed very low rates of inflammation in the sample, which is important for interpreting iron biomarker values. Limitations of the study included that this was a pilot study with a relatively small sample size and panel of analytes. Additionally, samples were drawn from a largely white, non-Hispanic population and the biorepository has limited maternal pre-pregnancy and pregnancy nutrient intake data, which limited analyses of risk factors for neonatal ID. Additionally, since analyses were run on previously collected serum samples, analytes that are measured in whole blood, such as hemoglobin and zinc protoporphyrin, could not be included in the present analysis. These analytes would have informed on anemia status and iron availability for erythropoiesis. Finally, the singletons were selected to have similar GA to the twins to reduce confounding by GA, but the included singletons may not be representative of the larger population of singleton neonates. Still, the ID prevalence data in the twin neonates is robust and our findings in total suggest an urgent need and directions for future research and intervention. 

## 5. Conclusions

ID is common in apparently healthy neonates and may inhibit cognitive development during the critical first months of life [6,7,40]. Broader newborn screening should be considered to identify neonates in need of supplementation earlier in infancy to improve cognitive and other outcomes. In parallel, research is needed to focus screening efforts by identifying characteristics of neonates at elevated risk of ID. Or, if strong maternal and pregnancy risk factors for neonatal ID can be identified, interventions could be initiated prenatally before fetal ID occurs to maximize protection of fetal and infant neurocognitive development. Concurrently, research is needed to better understand iron partitioning between mother, placenta, and fetus(es) to identify conditions under which fetal iron accrual is inadequate or is constrained by factors other than maternal iron availability. Iron is critical in the early months of life, both in utero and postnatally. Research and action are needed to protect vulnerable infants and improve equity in the nutritional foundation that infants receive to support their development.

## Figures and Tables

**Figure 1 nutrients-14-03854-f001:**
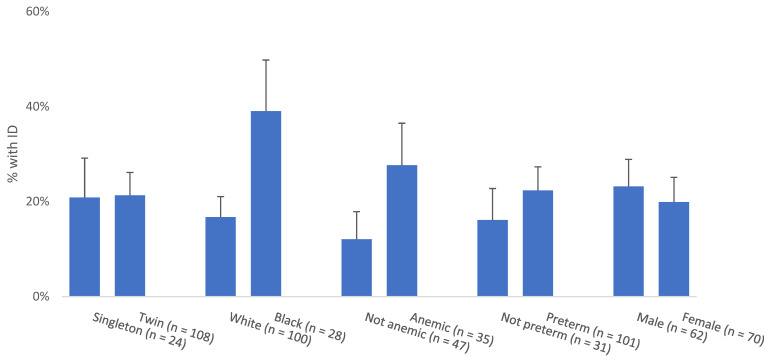
Prevalence of ID by selected maternal and infant characteristics. Bar height indicates the prevalence of ID within each level of the characteristic. Error bars indicate the standard error of the estimated prevalence. Prevalence and standard errors were estimated with mixed effect linear regression models with a random effect for twin pair. Prevalence differences did not reach statistical significance in mixed effects models. ID, iron deficiency.

**Figure 2 nutrients-14-03854-f002:**
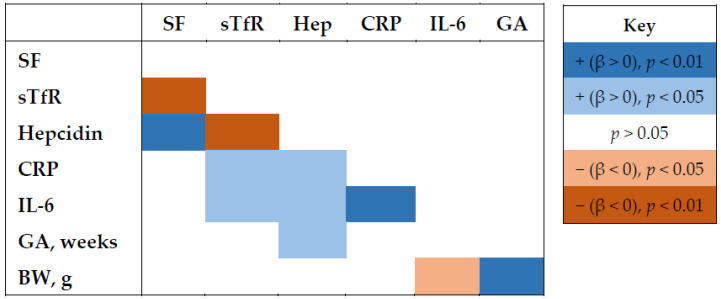
Associations among log-transformed iron and inflammation biomarkers, gestational age, and birth weight. Direction and strength of associations are indicated by color and intensity (see key). Values are from mixed effect models with random effects for twin pairs. *n* = 2 missing for IL-6. Abbreviations: BW, birthweight; CRP, c-reactive protein; GA, gestational age; IL-6, interleukin-6; SF, serum ferritin; sTfR, soluble transferrin receptor.

**Figure 3 nutrients-14-03854-f003:**
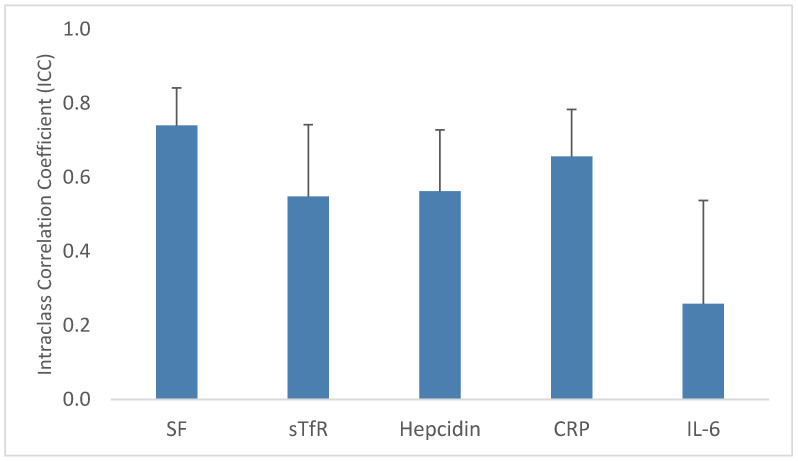
Neonatal iron biomarker intraclass correlation coefficients (ICCs). Bar height indicates the ICC for each biomarker calculated in mixed-effects linear regression models with a random effect for twin pair. Error bars are 95% confidence interval upper bounds from the same model. Abbreviations: CRP, c-reactive protein; IL-6, interleukin-6; SF, serum ferritin; sTfR, soluble transferrin receptor.

**Table 1 nutrients-14-03854-t001:** Maternal, pregnancy and infant characteristics in sampled biorepository participants.

Characteristic	Twin Pregnancies				Singleton Pregnancies				*p*-Value ^1^
	*n*	%	Mean	SD	*n*	%	Mean	SD	
Maternal and Pregnancy									
Race	54				24				0.253
White	42	77.8			16	66.7			
Black	10	18.5			8	33.3			
Other	2	3.7			0	0.0			
Hispanic ethnicity	54				24				0.340
No	52	96.3			24	100			
Yes	2	3.7			0	0			
Maternal age, year	54		28.9	6.1	24		27.9	6.4	0.497
Gravidity	54				24				0.023
1	26	48.1			5	20.8			
2+	28	51.9			19	79.2			
Parity	54				24				0.096
0	29	53.7			8	33.3			
1+	25	46.3			16	66.7			
BMI	33		35.7	8.2	12		37.8	8.2	0.457
Anemia (Hb <11 g/dL)	35				12				0.207
No	19	54.3			9	75.0			
Yes	16	45.7			3	25.0			
Preeclampsia	54				24				0.766
No	40	74.1			17	70.8			
Yes	14	25.9			7	29.2			
Gestational and/or chronic hypertension	54				24				0.143
No	52	96.3			21	87.5			
Yes	2	3.7			3	12.5			
Twin type	54								
Di-di	47	87.0							
Mono-di	7	13.0							
Gestational age (GA), weeks	54		34.9	2.2	24		35.2	2.9	0.529
Preterm (GA <37 weeks)	54				24				0.510
No	12	22.2			7	29.2			
Yes	42	77.8			17	70.8			
Delivery context	54				24				0.753
Term delivery	12	22.2			7	29.2			
Spontaneous PTB	20	37.0			9	37.5			
Provider-initiated PTB	22	40.7			8	33.3			
Delivery type	54				24				0.441
Vaginal	23	42.6			8	33.3			
C-section	31	57.4			16	66.7			
Neonatal									
Birth weight (BW), g	108		2239.1	477.6	24		2639.0	749.7	0.001
LBW (BW <2500 g)	108				24				0.045
No	39	36.1			14	58.3			
Yes	69	63.9			10	41.7			
VLBW (BW <1500 g)	108				24				0.887
No	98	90.7			22	91.7			
Yes	10	9.3			2	8.3			
Sample type	108				24				0.409
Serum	105	97.2			24	100			
Plasma	3	2.8			0	0			

**^1^***p*-value from χ2 test for dichotomous characteristics and *t*-test for continuous characteristics. Abbreviations: BMI, body mass index; BW, birth weight; GA, gestational age; Hb, hemoglobin; LBW, low birthweight; PTB, preterm birth; SD, standard deviation; VLBW, very low birthweight.

**Table 2 nutrients-14-03854-t002:** Summary of iron biomarker values in neonates from twin and singleton pregnancies.

Biomarker		Twins			Singletons			
	*n*	%	Mean	SE	*n*	%	Mean	SE
SF, ng/mL	108		176.33	15.36	24		158.46	22.44
ID (SF <76 ng/mL)	23	21.3			5	20.8		
sTfR, ng/mL	108		2501.55	98.98	24		3527.85	404.99
Hepcidin, ng/mL	108		14.95	1.65	24		13.68	2.44
CRP, ng/ml	108		116.46	46.14	24		66.74	12.75
Elevated (CRP ≥95th %ile)	6	5.6			1	4.2		
IL-6, pg/mL	107		13.58	7.22	24		2.96	0.87
Elevated (IL-6 ≥100 pg/mL)	3	2.8			0	0		

Abbreviations: CRP, c-reactive protein; ID, iron deficient; IL-6, interleukin-6; SE, standard error; SF, serum ferritin; sTfR, soluble transferrin receptor.

**Table 3 nutrients-14-03854-t003:** Maternal, pregnancy and infant characteristics associated with cord blood serum ferritin: multivariable regression model.

Characteristic	Β ^1^	95% CI
Race		
White		
Black	−0.53	(−0.94, −0.12)
Other	0.21	(−0.84, 1.27)
Twin status		
Singleton		
Twin	0.05	(−0.33, 0.44)
Preterm (<37 weeks)		
No		
Yes	−0.54	(−0.94, −0.15)
Sex		
Male		
Female	0.32	(0.11, 0.52)

^1^ Coefficients and 95% confidence intervals are from mixed effect models with random effects for twin pairs. Negative β values suggest an inverse relationship of the variable with cord blood serum ferritin. Confidence intervals (CI) that do not contain zero indicate consistency with a non-zero association between the variable and cord blood serum ferritin (i.e., statistical significance).

**Table 4 nutrients-14-03854-t004:** Predictors of intrauterine variability in log-transformed iron biomarkers.

	Percent of Intrauterine Variability Explained		
	SF	sTfR	Hepcidin
Hepcidin	24.5	11.6	
CRP	0.2	−1.1	−6.6
IL-6	1.8	−1.3	−3.2
Birth weight	−0.6	0.5	3.2
Infant sex	−2.7	−1.2	−4.2
Maternal race	5.0	5.3	1.1
Maternal anemia	12.2	−2.5	−1.1
Maternal BMI	9.8	−6.3	0.1

Values are the percent change in within-cluster residual variance comparing the empty model to one with the specified variable as an independent variable. Abbreviations: BMI, body mass index; CRP, c-reactive protein; IL-6, interleukin-6.

## Data Availability

The data presented in this study are available on request from the corresponding author. The data are not publicly available due to patient privacy.

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
