# Peer review of "Prevalence of and Risk Factors for Iron Deficiency in Twin and Singleton Newborns"

_nutrients, 2022, doi:10.3390/nu14183854_

Round 1

Reviewer 1 Report

This is an interesting and informative manuscript addressing the prevalence of and risk factors for iron deficiency anemia in twin and singleton newborns. The authors found that the prevalence of ID among twins and singleton newborns were similar, 21% vs 20%. Gestational age at birth, maternal race and infect sex were related to serum ferritin levels. Maternal anemia was not associated to neonatal iron biomarkers. Neonates of black mothers had higher prevalence of ID than neonates of white mothers. In addition, neonates of born preterm or to mothers with anemia had higher prevalence of ID compared to those whose mothers did not have anemia and who were full term. Maternal race, preterm birth and infant sex independently predicted serum ferritin. ID was highly concordant within twin pairs. Cord blood hepcidin explained 25% intrauterine variability in serum ferritin and 15% intrauterine variability in soluble transferrin receptor.

In the discussion, the authors recommend early infancy screening for ID and expanded iron suplementation in order to protect iron status and improve cognitive development in iron deficient neonates.

The authors site many relevant publications. One authored by Jie Shao, Blair Richards, Niko Kaciroti, Bingquan Zhu, Katy M Clark and Betsy Lozoff found that Indicators of iron status at birth, postnatal iron needs, and iron sources independently related to iron status at 9 months. Sex was an additional factor. They recommend that public health policies to identify and protect infants at increased risk of ID should be prioritized.

Author Response

We appreciate Reviewer 1’s detailed reading and favorable review of our manuscript.

Reviewer 2 Report

In this article, the authors showed their efforts on exploring the prevalence and risk factor for ID. There are several major concerns list below.

1.     Line 56-61 The authors mentioned race and ethnicity are associated with ID burden in the US. I just wonder if other characteristics such as income, dietary structure can also be influential factors of ID burden.

2.     Line 96-97 If the protocol of the immunoassay kits allowed to be run using plasma, I recommend the authors use a coefficient to reduce concentration difference between serum and plasma. If not, the 3 samples should not be used in further analysis.

3.     Line 114 The mono-di and di-di should be define at the first present in Line 83.

4.     Low birth weight and preterm birth are often associated with pregnancy comorbidities, did the authors screen the study subjects? Moreover, what is the include and exclude criteria of the current research?

5.     The method of statistical analysis was too brief and the necessary parameters and covariates of each model should be supplemented. The current description was hard for further researchers to recurrence the result.

6.     The authors seemed use twins and singletons in statistical models with equal power. Which may lead the characteristics of mothers who gave birth to twins have 2-time power than singleton mothers. I recommend the authors only use one of each pair of twins in models to solve this issue.

7.     Since the authors declared the current research aimed to evaluate fetal iron status at birth in twin versus singletons, the sample sizes varied considerably and was contrary to the normal ratio. The authors should give a reasonable explanation of the scientific nature of the tobacco and alcohol population

8.     Figure 2, What is the difference between block colored red and blue? I recommend the authors show more detail data of their results in figures.

Round 2

Reviewer 2 Report

I am satisfied with the revisions made by the authors and congratulate them on a good article.